# ONCOBREAST-TEST Is a Quick Diagnostic, Prognostic and Predictive Method of Response to Systemic Treatment

**DOI:** 10.3390/cancers16010120

**Published:** 2023-12-26

**Authors:** Anna Tankiewicz-Kwedlo, Tomasz Lobacz, Leszek Kozlowski, Bogumila Czartoryska-Arlukowicz, Mariusz Koda, Krystyna Pawlak, Robert Czarnomysy, Magdalena Joanna Borkowska, Dariusz Pawlak

**Affiliations:** 1Department of Pharmacodynamics, Medical University of Bialystok, Mickiewicza 2C, 15-222 Bialystok, Poland; dariusz.pawlak@umb.edu.pl; 2M. Skłodowska-Curie Bialystok Oncology Center, Ogrodowa 12, 15-027 Bialystok, Poland; tomek.lobacz91@gmail.com (T.L.); lkozlowski@onkologia.bialystok.pl (L.K.); barlukowicz@poczta.onet.pl (B.C.-A.); magdaborkowska74@gmail.com (M.J.B.); 3Department of General Pathomorphology, Medical University of Bialystok, ul. Waszyngtona 13, 15-269 Bialystok, Poland; kodamar2@gmail.com; 4Department of Monitored Pharmacotherapy, Medical University of Bialystok, Mickiewicza 2C, 15-222 Bialystok, Poland; krystyna.pawlak@umb.edu.pl; 5Department of Synthesis and Technology of Drugs, Medical University of Bialystok, Kilinskiego 1, 15-089 Bialystok, Poland; robert.czarnomysy@umb.edu.pl

**Keywords:** breast cancer, chemotherapy, chemosensitivity, personalized therapy

## Abstract

**Simple Summary:**

Despite the rapid development of medicine, the appropriate selection of anticancer drugs in breast cancer is still not optimized. To address personalized medicine, we propose a simple diagnostic and therapeutic pathway, called the ONCOBREAST-TEST, which aims to select drugs against which a patient’s cancer cells, taken during a biopsy, show the greatest sensitivity. Our research showed that simple, inexpensive, and quick tests such as the assessment of cell viability, morphology, and lactate dehydrogenase activity are sufficient tools for selecting drugs with the highest anticancer efficacy. The conducted in vitro studies showed a high probability of accurate drug selection based on the proposed testing techniques.

**Abstract:**

ONCOBREAST-TEST is a diagnostic and therapeutic procedure that is part of the comprehensive care of a patient with breast cancer.: Chemosensitivity of cancer cells was assessed using the MTT test, morphological assessment of cells, LDH activity in the culture medium, and flow cytometry technique (apoptosis, proliferation, CD24, CD44, GATA3, cytokeratin, Ki-67). Diagnostic tools included panels of simple tests which could be used to accurately predict the chemosensitivity of tumor cells previously isolated from a patient, even before actual chemotherapy. The proposed procedure allows for a simple (based on MTT results, cell morphology, LDH concentration), minimally invasive, quick, and accurate assessment of the sensitivity of breast cancer cells to the drugs used and, to select the most effective treatment plan as part of personalized therapy. In a patient with NOS G3, the most promising therapy will be docetaxel with cyclophosphamide and in the case of a patient with NOS G1, paclitaxel alone and in combination with trastuzumab. The implementation of such a procedure would undoubtedly increase the effectiveness of chemotherapy, reduce side effects by excluding drugs that are ineffective before using them, protect the patient’s health, and shorten the treatment time, bringing economic and social benefits.

## 1. Introduction

According to the WHO, breast cancer is the most commonly diagnosed cancer in the world—diagnosed in 12.5% of all new annual cancer cases worldwide [1]. The total number of breast cancer cases in the world has increased by almost 100% in the last 20 years. It was estimated that breast cancer accounted for 13.3% of all new cancer cases diagnosed in the EU-27 in 2020. Therefore, it is considered the most common neoplasm [2]. Breast cancer was estimated to account for 28.7% of all new cancers in women in 2020, 2.3 million women were diagnosed with breast cancer worldwide, and 685,000 deaths were reported [3].

In Poland, a steady increase in morbidity continues despite additional financial outlays and medical development, as well as increasing social awareness in the field of cancer prevention. The death rate among Polish women suffering from breast cancer grows every year. Therefore, it is extremely important to develop an innovative diagnostic and therapeutic approach that allows for quick diagnostics and targeted chemotherapy as part of personalized medicine [4]. Despite thousands of studies, progress in diagnostic tests—including molecular and imaging tests—and the introduction of new chemotherapeutic options, selecting the optimal therapy for each patient is a tedious task as it is often ambiguous.

Dynamic progress in cancer diagnostics and learning about their biology has enabled the development of personalized treatment. Modern oncology can offer a targeted course of treatment tailored to the subtype of a given tumor. It is mainly based on multigene tests [4]. The results of these studies provide a lot of valuable information on the selected genes of the patient, but they make it impossible to monitor the response of neoplastic cells to the applied treatment.

The subject of the invention is the ONCOBREAST-TEST—a simple, fast, and inexpensive diagnostic and therapeutic path in the comprehensive care of a patient with breast cancer. The developed and proposed diagnostic procedure and the assessment of the sensitivity of neoplastic cells obtained directly from the tumor ensure quick treatment and a fully personalized approach to the oncological patient.

## 2. Materials and Methods

### 2.1. Study Population

Eight female patients with diagnosed breast cancer, who had not received any therapy so far, recruited between March 2022 and November 2022, were examined. The mean age of the patients was 62.61 ± 2.67 years. The patients were hospitalized at The Maria Sklodowska-Curie Oncology Center in Bialystok. These patients did not receive anticancer treatment, but due to age and concomitant diseases, such as hypertension, they received convertase inhibitors and β-blockers.

Ethical approval for this study was obtained from the Ethics Committee of the Medical University of Bialystok, Poland (APK.002.28.2021). All procedures performed during the study were in accordance with the Declaration of Helsinki. Participation in the study was voluntary and all participants provided a written notification agreement. The qualified participants received detailed information and clarification of all procedures before starting the test.

#### Inclusion and Exclusion Criteria

Women aged >18 years with newly diagnosed breast cancer, not undergoing any therapy, and living permanently in Poland were included in the study. The exclusion criteria were women with a history of mental disorders, and those who refused to participate in the study.

### 2.2. Core Needle Biopsy of the Breast

The first stage of research was a core-needle biopsy. After locating the focal lesion with a linear probe, the lesion area was disinfected and then infiltrated with 1% lidocaine solution (3–5 mL). Ropivacaine was used in cases of lidocaine allergy. The skin was incised with scalpel blade no. 11, then, after an ultrasound examination, biopsy needles (14 gauge) were inserted into the focal lesion of the breast and the material was collected for histopathological examination. One piece of tissue was placed in a vessel with a sterile medium and immediately transferred to a cell culture laboratory to obtain tumor cells. The remaining tissue fragments were placed in a 10% buffered formalin solution and transferred to the Department of Pathology for routine histopathological diagnosis. A sterile wound dressing was applied to the patient, followed by a compression dressing on the chest, which was left for approximately 12 h.

### 2.3. Immunohistochemical Staining

After collection, the tissue material was fixed in 10% buffered formalin solution, pH 7.2–7.4, at room temperature (20–25 °C), in a volume 10 times the volume of the sample tissue. The curing time was 24 h.

The referral, in addition to the patient’s data, contained the exact date and time of tissue collection and placement in the fixative. After the fixation step, the sterile biopsy vessel was fixed. For this purpose, they were transferred to disposable plastic histopathological cassettes, clearly and permanently marked, enabling the identification of the patient and the performed examination. The cassettes containing the tissue material were then placed in an automated tissue processor, allowing the material to be introduced into the paraffin. The time, temperature, and pressure of the reagents were controlled in a control device. After the tissue processor was finished, the biopsy samples were transferred to metal molds and embedded in paraffin blocks using a paraffin machine. Then, the blocks were cooled and cut using microtomes, enabling the cutting of a section with a thickness of approx. 5 µm, which was placed on a microscope slide.

For routine histopathological diagnosis, sections were stained with hematoxylin and eosin.

The histopathological diagnosis was performed using an optical microscope with a magnification of up to 400×. In the case of a malignancy diagnosis, immunohistochemical tests were performed to assess predictors, such as the estrogen receptor, progesterone receptor, HER2 receptor and proliferative marker Ki-67. The immunohistochemical examination included the following steps: exposure of antigenic determinants, blocking of non-specific reactions, administration of primary antibodies, and the use of detection systems using diaminobenzidine (DAB).

In the diagnosis of breast cancer, antibodies in the form of ready-to-use solutions available on the market (Roche, Basel, Switzerland) were used. A positive control was run for each reaction and placed on the same slide as the test section. Immunohistochemistry was performed using automated systems (Benchmark, Ventana, Tucson, AZ, USA). After the reaction, the stained and dehydrated slides were permanently covered with a coverslip and subjected to microscopic examination.

### 2.4. Obtaining Cancer Cells

Immediately after collection, the third biopsy was placed in a DMEM medium, heated to 37 °C, supplemented with 10% serum and penicillin (100 IU/mL), streptomycin (100 μg/mL) and amphotericin B (0.25 μg/mL). Within 60 min, the material was delivered to the laboratory and subjected to the tumor cell harvesting steps. Under sterile conditions, after removing the culture medium, the biopsy was washed with a sterile PBS solution. After cutting with a sterile scalpel, the tissue sections were placed on a sterile culture plate with IMDM medium supplemented with 10% serum, penicillin (100 IU/mL), streptomycin (100 μg/mL), and amphotericin B (0.25 μg/mL) supplemented with type IV collagenase (1 mg/mL). The plate was placed in an incubator at 37 °C in an atmosphere enriched with 5% carbon dioxide (CO_2_). After 8–12 h, the cell suspension was centrifuged at 1500 rpm at room temperature for 10 min. After decanting the supernatant, the cell pellet was resuspended in a sterile medium and transferred to a sterile culture plate with IMDM medium supplemented with 10% serum, penicillin (100 IU/mL), streptomycin (100 μg/mL), and amphotericin B (0.25 μg/mL). Cell proliferation capacity and morphology were assessed daily using an inverted microscope (Olympus, Tokyo, Japan). The culture medium was changed every three days. After reaching 80–90% confluence, the cells were passaged with 0.25% EDTA trypsin solution, and seeded into a new culture dish.

### 2.5. Verification of Cancer Cells

To verify the neoplastic cells by flow cytometry, the expression of CD24 and CD44 receptors was assessed. Breast cancer cells in the log phase of growth were used for flow cytometric analysis. To detach them from the substrate, the cells were treated with 0.25% trypsin, washed with PBS, then resuspended in 100 µL of PBS, followed by incubation with anti-CD44-BD Horizon™ BB515 and anti-CD24-BD Horizon™ BV510 antibodies at room temperature for 40 min. The samples were then washed with PBS and finally resuspended in 200 µL PBS. Flow cytometric analysis was performed on a BD FACSCanto II (BD Biosciences Systems). The ratio of CD44 and CD24 (CD44/CD24) expression in breast cancer cells from different patients was calculated from the percentage of CD44 and CD24 subpopulations positive by flow cytometric analysis.

In addition, the expression of Ki-67, cytokeratin, GATA3, and annexin V was assessed by flow cytometry in cells isolated from each patient.

Flow cytometric analysis was performed on breast cancer cells in log phase growth. After digestion with 0.25% trypsin and washing three times with PBS, the cells were resuspended in 100 µL of PBS and then stained with anti-Ki-67-BD Horizon™ BV421, anti-GATA3-PE-CyTM7 and anti-cytokeratin-Alexa Fluor 647 at room temperature for 40 min. The samples were then washed three times with PBS and finally resuspended in 200 µL PBS. The analysis was performed on a BD FACSCanto II flow cytometer (BD Biosciences Systems, San Jose, CA, USA). The ratio of CD44 and CD24 (CD44/CD24) and Ki-67, cytokeratin, and GATA3 expression in breast cancer cells were calculated from the percentage of Ki-67, GATA3, and cytokeratin positive subpopulations by flow cytometric analysis.

To assess cell chemosensitivity, apoptosis was investigated using the annexin V-FITC-labeled method of annexin V-FITC, which complexes with phosphatidylserine in the presence of calcium ions. This study was performed using the FITC Annexin V Apoptosis Detection Kit II (BD Biosciences, San Jose, CA, USA) and a BD FACSCanto II flow cytometer (BD Biosciences Systems, San Jose, CA, USA).

### 2.6. Assessment of Neoplastic Cell Chemosensitivity

Then, the tumor cells were incubated for 24 and 48 h with drugs used in patients with breast cancer according to the following schemes:Doxorubicin (4 μM) + 4-Hydroxycyclophosphamide (the active metabolite of cyclophosphamide) (1 μM)Cisplatin (20 μM)Paclitaxel (2 μM)Paclitaxel (2 μM) + Trastuzumab (0.7 μM)Docetaxel (1 μM)Docetaxel (1 μM) + Trastuzumab (0.7 μM)

After 24 and 48 h of incubation, viability was assessed by the Carmichael method, determining cell viability using tetrazolium salt (MTT) as previously described [5]. Absorbance readings were taken using a microplate reader (Bio-Tek Instruments, Winooski, VT, USA) at 570 nm. Cell viability was assessed against the 100% viable control.

To assess cell chemosensitivity, apoptosis was examined using the annexin V-FITC labeling method, which complexes with phosphatidylserine in the presence of calcium ions. This assay was performed using the FITC Annexin V Apoptosis Detection Kit II (BD Biosciences, Franklin Lakes, NJ, USA) and a BD FACSCanto II flow cytometer (BD Biosciences Systems, San Jose, CA, USA). Analysis of the results was performed using FACSDiva 6.1.3 software (BD Biosciences Systems, San Jose, CA, USA). To assess proliferation, the CFSE test was performed using a commercial kit from Biosciencesnces, which allowed for the detection of multiple generations of neoplastic cells by flow cytometry using the method described by Ganesanesam et al. [6].

#### Evaluation of Lactate Dehydrogenase (LDH) Activity in Culture Media

To confirm the effectiveness of the cytotoxic effect of the drugs used, the activity of lactate dehydrogenase (LDH) was assessed in the culture medium after the incubation of cells with chemotherapeutic agents. The activity of this enzyme increases when the cell loses its membrane integrity, which occurs when cells are chemically sensitive to an administered drug. For this purpose, after incubating the cells with individual chemotherapeutic agents according to the previous scheme, the culture medium was collected and stored at −80 °C until the assay. Then, 50 µL of 200 mM TRIS with a pH of 8, 50 µL of 50 mM lithium lactate, and 50 µL of a mixture of phenazine methosulfate, iodonitrotetrazolium chloride and nicotinamide adenine dinucleotide were added to 50 µL of the medium. Next, the mixture was incubated for 10 min and read at 490 nm using a reader for microplates (Bio-Tek Instruments, Winooski, VT, USA).

### 2.7. Statistics

All the calculations, statistical analyses, and graphs were performed using GraphPad Prism software version 10.1.0 (Irvine, CA, USA). Shapiro–Wilk tests were performed to determine whether continuous variables were normally distributed. Normally distributed data are presented as mean ± SD. In total, 2 patients (4 and 7) were compared using the non-parametric Mann–Whitney test. The significance level was *p* < 0.05.

## 3. Results

### 3.1. General Patient and Tumor Characteristics

The study included 8 patients with a diagnosis of invasive breast carcinoma (IBC), not otherwise specified (NOS) (Invasive ductal carcinoma) in different grades of histologic differentiation (G1-3). NOS G1 type occurred in 3 patients (37.5% of cases), NOS G2 type occurred in 2 patients (25% of cases) and NOS G3 type occurred in 3 patients (37.5% of cases). IBC NOS is the most common type of invasive breast carcinoma (75–80%). All IBC NOS arise from epithelial progenitor cells at the terminal duct lobular unit in the mammary gland. Histologic grade (G) is independent and one of the most important prognostic factors in breast carcinoma. Pathologists use the Nottingham, modified Bloom & Richardson Score for diagnosis [7]. Histologic grading is based on the microscopic analysis of tubule formation (1–3 points), nuclear pleomorphism (1–3 points) and mitotic count (1–3 points). Total score (after adding points for tubule formation, nuclear pleomorphism and mitotic count) divide all IBC NOS on: grade 1 (G1; 3–5 points), grade 2 (G2; 6–7 points) and grade 3 (G2; 8–9 points). Of the 8 patients, seven had ER+ tumors (87.50%) and seven had PR+ tumors (87.50%). Most cancers were HER2+ (n = 5; 62.50%). Table 1 presents the characteristics of the patients included in the study.

### 3.2. Isolation of Neoplastic Cells from Tumor Stroma

Tumor cells were isolated from each biopsy. Under the influence of collagenase, after 3–6 h of incubation, neoplastic cells, which initially remained bound in the tumor tissue, were visualized. As the duration of collagenase action increased, these cells were released from the stroma and appeared in the medium. After removing the enzyme from the medium, a proliferating cell colony appeared within a week, but this time varied depending on tumor type and patient age (Figure 1A–D).

### 3.3. Extensive Research Panel

#### 3.3.1. Confirmation of Cellular Heterogeneity

In the first stage of the study, two patients (marked in red in Table 1) were randomly selected for detailed analysis. Since a growing tumor in a patient’s body is a heterogeneous creation, it is important to preserve such an environment as reliably as possible in in vitro studies. To evaluate the analysis of these conditions, samples were subjected to cytometric analysis. We observed that three cell lines were present in both materials from patients 4 and 7. The results show that cellular heterogeneity was preserved in both cases and therefore all in vitro tests were carried out under conditions as close as possible to those in the body of an oncology patient (Figure 2).

#### 3.3.2. Evaluation of Extracellular Marker Expression

It is postulated that the group of CD44 and CD24 receptors are favorable prognostic markers in breast cancer. CD24 protein expression is considered a poor prognostic marker in hormone receptor-positive breast cancer, in contrast to CD44 expression, which is a good prognostic marker in the group of patients without hormone receptors [8]. Using the appropriate antibodies and the flow cytometry method, cancer stem cells with the CD44(+) CD24(+) phenotype were identified in each material. This clearly proves the presence of breast cancer cells in the material obtained from the biopsies. Both patients 4 and 7 had comparable CD24 expression levels of 94.86 ± 6.37% (95% CI = 91.33; 98.39%) and 96.75 ± 0.95% (95% CI = 95.95; 97.55%), respectively (Figure 3, Appendix A). The expression of the CD44 receptor in patient 4 was 89.21 ± 7.90% (95% CI = 84.65; 93.77%), and in patient 7 it was 97.9 ± 1.80% (95% CI = 96.39; 99.41%) (Figure 3). A comparison of the CD44/24 expression ratio showed no significant differences between the patients included in the study. The ratio was 1.09 ± 0.17 (95% CI = 0.98; 1.19) in patient 4 and 0.99 ± 0.03 (95% CI = 0.97; 1.00) in patient 7 (Figure 3, Appendix A). Based on cytometric analysis, it is clear that breast cancer cells were present in the material collected from the tumor.

#### 3.3.3. Evaluation of Intracellular Marker Expression

In cells obtained from the biopsy, GATA-3 expression in patient 4 was statistically significantly higher (*p* = 0.018) and amounted to 52.24 ± 15.6% (95% CI= 39.20; 65.28) compared to patient 7 (32.24 ± 12.48%) (95% CI = 20.71; 43.78) (Figure 4A,B, Appendix A). According to the literature, this parameter is associated with favorable pathological features of breast cancer, including the absence of lymph node metastases and estrogen receptor (ER) positive status. GATA-3 was also shown to be an independent prognostic marker, with low expression predicting breast cancer recurrence [9].

The second intracellular marker assessed in vitro was cytokeratin—an insoluble structural protein of epithelial cells. This marker is associated with the proliferative capacity of tumor cells. Its presence in the blood is an unfavorable factor indicating rapid disease recurrence [10]. Cytokeratin expression in patient 4 was statistically significantly higher (*p* = 0.013) and amounted to 53.16 ± 15.96% (95% CI = 38.40; 67.92) compared to patient 7, in whom it was 31.27 ± 11.58% (95% CI = 20.56; 41.99) (Figure 4C,D, Appendix A).

Ki-67 expression differed statistically significantly (*p* = 0.026) between patients and was 53.1 ± 15.94% (95% CI = 38.31; 67.80) in patient 4 and 32.99 ± 14.41% (95% CI = 19.66; 46.31) in patient 7 (Figure 4E,F, Appendix A). The obtained results are similar to the immunohistochemical results performed three months earlier and were 70% and 15%, respectively (Figure 4G, Appendix A). The conducted observations once again confirm the presence of tumor cells in the performed in vitro tests, and furthermore, based on the similarity of the results obtained from cytometric analysis and immunohistochemistry, indicate the in vitro stability of the cell lines isolated from the tumors for three months.

### 3.4. Evaluation of Cancer Cell Chemosensitivity

#### 3.4.1. Cell Viability Testing

Cells obtained from biopsies were incubated with drugs for 48 h. There was a statistically significant reduction in cell viability from patient 4 after incubation with all drugs at each time interval. However, after 24 h of incubation, this effect was most strongly observed after exposing cells to Doc (72.40 ± 1.15% (95% CI = 71.19; 73.61) vs. 100% (95% CI = 90.71; 109.30) in control; *p* < 0.0001), Dox + Cyclo (63.24 ± 0.7% (95% CI = 62.53; 63.94) vs. 100% in control; *p* < 0.0001), Doc + Tras (65.87 ± 0.96% (95% CI = 65.07; 66.68) vs. 100% in control; *p* < 0.0001), Pac + Tras (77.75 ± 3.87% (95% CI = 74.51; 80.99) vs. 100% in control; *p* < 0.0001), and Pac (67.78 ± 1.41% (95% CI = 66.30; 69.25) vs. 100% in control; *p* < 0.0001) (Figure 5A, Appendix A).

In the case of patient 7, after 24 h incubation, the decrease in viability was most strongly observed after exposing cells to Doc + Tras (71.12 ± 9.77% (95% CI = 62.09; 80.15) vs. 100% (95% CI = 90.77; 109.23) in the control; *p* < 0.0001), Doc (79.99 ± 2.31% (95% CI = 77.57; 82.41) vs. 100% in control; *p* < 0.0001), Dox + Cyclo (63.94 ± 0.83% (95% CI = 61.44; 63.10) vs. 100% in control; *p* < 0.0001), Pac + Tras (67.33 ± 1.84% (95% CI = 65.79; 68.87) vs. 100% in control; *p* < 0.0001), and Pac (67.18 ± 1.80% (95% CI = 65.29; 69.07) vs. 100% in control; *p* < 0.0001) (Figure 5B, Appendix A).

#### 3.4.2. Assessing the Effects of Drugs on Cancer Cell Morphology

Evaluation of the morphology of the drug-treated tumor cells under visible light provides only a preliminary opportunity for the selection of cytotoxic drugs. However, supported by the results of other studies, it provides an opportunity to select drugs that are most beneficial to the oncology patient. The incubation of the cells of patients 4 (Figure 6A) and 7 (Figure 6B) resulted in the loss of adherent properties and cell-to-cell contact after exposure to paclitaxel (Pac), paclitaxel with trastuzumab (Pac + Tras), docetaxel (Doc), and docetaxel with trastuzumab (Doc + Tras). In contrast, cisplatin (Cis) or doxorubicin with cyclophosphamide (Dox + Cyclo) did not disrupt tumor cell morphology.

#### 3.4.3. Assessment of Cell Proliferation

The cytometric test showed differences in the proliferation of cells treated with several chemotherapeutics. This made it possible to exclude from the patient’s future therapy drugs that did not inhibit or slow down proliferation; on the other hand, it indicated the drugs to which the patient’s tumor cells showed sensitivity.

In the case of patient 4, only the fourth cell generation was present in cells not exposed to the drugs, clearly indicating persistent proliferation. All the drugs used in the study inhibited cells at the third generation, demonstrating their potential efficacy in oncology treatment. However, the strongest proliferation inhibition effect was observed after the exposure of cells to doxorubicin with cyclophosphamide (Dox + Cyclo) (60.47 ± 6.15%; 95% CI = 45.18; 75.75 in 1st generation, 77.30 ± 6.50; 95% CI = 61.15; 93.45 in 2nd generation and 25.23 ± 2.57%; 95% CI = 18.84; 31.61 in 3rd generation of control group) (Figure 7A, Appendix A). In patient 7, the fourth generation appeared after the cells were exposed to docetaxel with trastuzumab (Doc + Tras) (478.80 ± 49.97%; CI = 354.60; 602.90 in the first generation, 283.00 ± 30.98; 95% CI = 206.00; 359.90 in the second generation, 407.10 ± 38.96%; 95% CI = 310.40; 503.90 in the third generation and 450.00 ± 85.44%; 95% CI = 237.80; 662.20 in the fourth generation of the control group), while in the case of the control group or the other chemotherapeutics, proliferation slowed on the third generation. This indicates that in this patient, the use of an immunomodulatory drug like trastuzumab enhances tumor cell proliferation. Also, the addition of trastuzumab (Pac + Tras) to paclitaxel (423.84 ± 52.86%; CI = 293.00; 555.60 in the first generation, 318.88 ± 34.59; 95% CI = 232.90; 404.80 in the second generation, 604.50 ± 52.72%; 95% CI = 473.60; 735.50 in the third generation of the control group) intensifies this effect, which is consistent with the assumption (Figure 7B, Appendix A).

#### 3.4.4. Evaluation of Apoptosis

After 24 h of incubation of breast cancer cells with chemotherapeutics, their effect on apoptosis induction was evaluated. It was shown that all the drugs used lead to programmed cell death as a result of their action. The strongest apoptosis was observed after the incubation of cells with paclitaxel (Pac) in patient 4. In this group, early apoptosis was 427.40 ± 26.98% (95% CI = 360.40; 494.50) of the control, late apoptosis was 2733.0 ± 177.60% (95% CI = 2291; 3174) of the control, necrosis was 66.23 ± 5.56% (95% CI = 52.42; 80.04) of the control, and viable cells accounted for 78.65 ± 6.57% (95% CI = 62.33; 94.97) of the control. In addition to paclitaxel, doxorubicin with cyclophosphamide (Dox + Cyclo) (early apoptosis 229.3 ± 17.79% (95% CI = 185.20; 273.50) of control, late apoptosis 1671 ± 143.7% (95% CI = 1314; 2028) of control, necrosis 166.7 ± 17.5% (95% CI = 123.20; 210.10) of control, viable cells 90.39 ± 3.857% (95% CI = 80.80; 99.97) of control), paclitaxel with trastuzumab (Pac + Tras) (early apoptosis 229.3 ± 17.79% (95% CI = 185.20; 273.50) of controls, late apoptosis 1880.0 ± 108.6% (95% CI = 1610; 2149) of controls, necrosis 132.0 ± 10.54% (95% CI = 105.8; 158.2) of controls, viable cells 89.11 ± 6.59% (95% CI = 72.75; 105.50) of controls), and docetaxel with trastuzumab (Doc + Tras) (early apoptosis 240.10 ± 19.04% (95% CI = 192.8; 287.4) of controls, late apoptosis 1667.0 ± 78.89% (95% CI = 1471; 1863) of controls, necrosis 166.7 ± 9.02% (95% CI = 144.30; 189.10) of controls, viable cells 90.37 ± 4.51% (95% CI = 79.17; 101.6) of controls). In contrast, due to the high percentage of viable cells (94.65 ± 4.15% (95% CI = 83.33; 105.0) of controls), patient no. 4 should not be treated with cisplatin (Cis) (early apoptosis 193.0 ± 11% (95% CI = 165.60; 220.30) of controls, late apoptosis 602.7 ± 28.10% (95% CI = 532.90; 672.50) of controls, necrosis 134.3 ± 11.06% (95% CI = 106.90; 161.80) of controls) or docetaxel (Doc) alone (early apoptosis 171.6 ± 16.96% (95% CI = 129.50; 213.80) of controls, late apoptosis 444.0 ± 31.88% (95% CI = 365.10; 523.50) of controls, necrosis 200.0 ± 10.00% (95% CI = 175.20; 224.80) of controls) (Figure 8A, Appendix A).

In contrast, 24 h incubation of breast cancer cells obtained from a biopsy from patient 7 showed the greatest enhancement of apoptosis by doxorubicin with cyclophosphamide (Dox + Cyclo) (early apoptosis 135.1 ± 12.52% (95% CI = 104.00; 166.20) of control, late apoptosis 153.4 ± 10.61% (95% CI = 127.10; 179.70) of control) and necrosis (199.30 ± 19.01% (95% CI = 152.10; 246.60) of control). Viable cells were only 48.20 ± 5.11% (95% CI = 35.51; 60.88) of the control. In this study, additional drugs such as paclitaxel (Pac) (early apoptosis 153.7 ± 13.19% (95% CI = 120.90; 186.50) of control, late apoptosis 143.4 ± 11.02% (95% CI = 116.0; 170.80) of control, necrosis 103.7 ± 11.93% (95% CI = 74.03; 133.30) of control, viable cells 47.47 ± 5.07% (95% CI = 34.88; 60.05) of control), paclitaxel with trastuzumab (Pac + Tras) (early apoptosis 154.4 ± 12.50% (95% CI = 123.30; 185.40) of control, late apoptosis 140.4 ± 11.50% (95% CI = 111.80; 169.0) of control, necrosis 101.3 ± 13.05% (95% CI = 68.91; 133.80) of control, viable cells 47.20 ± 5.63% (95% CI = 33.21; 61.19) of controls), docetaxel (Doc) (early apoptosis 152.40 ± 12.59% (95% CI = 121.10; 183.60) of controls, late apoptosis 122.2 ± 15.07% (95% CI = 84.72; 159.60) of controls, necrosis 101.0 ± 12.53% (95% CI = 69.87; 132.10) of controls, live cells 58.00 ± 6.56% (95% CI = 41.71; 74.29) of controls), as well as doxorubicin with cyclophosphamide (Dox + Cyclo) (early apoptosis 135.10 ± 12.52% (95% CI = 104.00; 166.20) of controls, late apoptosis 153.4 ± 10.61% (95% CI = 127.10; 179.70) of controls, necrosis 199.30 ± 19.01% (95% CI = 152.10; 246.60) of controls, live cells 48.20 ± 5.11% (95% CI = 35.51; 60.88) of controls) was observed. Due to the high percentage of viable cells (73.91 ± 5.54% (95% CI = 60.14; 87.69) of controls), this patient should not be treated with cisplatin (Cis), despite increased apoptosis (early apoptosis 121.8 ± 12.95% (95% CI = 89.62; 154.00) of controls, late apoptosis 127.6 ± 9.31% (95% CI = 104.40; 150.70) of controls) and necrosis (201.7 ± 21.55% (95% CI = 148.10; 255.20) of controls) (Figure 8B, Appendix A).

#### 3.4.5. Evaluation of LDH Activity in Culture Medium

Lactate dehydrogenase (LDH) is an intracellular enzyme that, under physiological conditions, does not escape outside the cell. Due to damage to the continuity of the cell membrane, when a cytotoxic agent is acted upon the activity of this enzyme in the extracellular environment increases.

In our experiment, we evaluated the activity of this enzyme in the extracellular environment, i.e., the culture medium, after 24 h of incubation of cancer cells with chemotherapeutic agents. In patient 4, the highest LDH activity was observed after exposure of cells to paclitaxel (Pac) (85.23 ± 4.0%; (95% CI = 75.30; 95.17) of controls). Other drugs with potentially high cytotoxicity include doxorubicin with cyclophosphamide (Dox + Cyclo) (68.34 ± 9.77%; (95% CI = 44.08; 92.60) of controls), paclitaxel with trastuzumab (Pac + Tras) (72.15 ± 15.90%; (95% CI = 46.86; 97.44) of controls), and docetaxel with trastuzumab (Doc + Tras) (61.88 ± 14.21%; (95% CI = 39.27; 84.49) of controls) (Figure 9A).

In contrast, in patient 7, the greatest cytotoxicity was observed after the incubation of cells with paclitaxel and trastuzumab (Pac + Tras) (85.21 ± 13.92%; (95% CI = 50.64; 119.80) of controls) and cisplatin alone (Cis) (77.65 ± 15.45%; (95% CI = −61.19; 216.50) of controls). Another therapeutic option, but with a slightly weaker cytotoxic effect, is the use of paclitaxel (Pac) (71.85 ± 8.39%; (95% CI = 51.01; 92.70) of controls) as monotherapy (Figure 9B).

The results of determining LDH activity in the culture medium support the selection of drugs with the greatest cytotoxicity against each patient’s tumor cells.

Based on this study, it can be concluded that the MTT assay, morphological evaluation of tumor cells and LDH activity in the culture medium after exposure of tumor cells to drugs is sufficient to select the most effective drugs. This approach will reduce the test time and cost.

### 3.5. Proposed Procedure

Based on this research, we developed an abbreviated diagnostic and therapeutic pathway, which we applied to six other patients with breast cancer. The purpose of these studies was to try to answer whether having quick, simple and inexpensive tests could select drugs with the potentially most potent anti-tumor effect.

Based on the MTT test (Table 2), morphological analysis (Table 3) and the assessment of LDH activity (Table 4), chemotherapy with the highest probability of success can be selected (Table 5).

Based on these tests, it is possible to select the drug/drugs to which the patient’s cancer cells are the most sensitive. The combined summary of the results obtained in the MTT test, microscopic evaluation, and LDH activity evaluation creates a real chance to select drugs with the highest anticancer activity (Table 5).

## 4. Discussion

The developed and proposed diagnostic procedure and assessment of the sensitivity of cancer cells obtained directly from the tumor provides targeted, rapid treatment based on a fully personalized approach to the oncological patient. So far, no simple, inexpensive, and effective diagnostic and therapeutic test has been developed to assess the chemosensitivity of cancer cells to chemotherapeutic agents before administering basic treatment to a specific oncological patient. Currently, the choice of chemotherapeutic treatment is based on the assessment of the patient’s clinical condition, the result of histopathological examination of the material collected during the biopsy, and molecular tests assessing predispositions, which only allow for estimating the prognosis for patients. Existing methods do not allow the assessment of the response of cells isolated from the patient to the proposed treatment. The scheme we present undoubtedly has great potential for discovery, diagnosis, and therapy. It enables the use of targeted chemotherapy with a high probability of effectiveness, guaranteeing the patient’s cure while limiting the side effects resulting from the use of inappropriate chemotherapy. Another advantage of such a procedure is shortening the total treatment time of an oncological patient through the use of chemotherapeutic agents to which the patient’s cancer cells are sensitive from the very first stages of treatment.

The currently widely used commercial cell lines do not reflect the histopathological status of the primary tumor because they represent a cell monoculture devoid of cellular stroma. Two popular in vitro breast cancer lines—MCF-7 and MDA-MB-123—were not isolated directly from the primary tumor, but from pleural effusion more than three years after mastectomy and radiotherapy/hormone therapy (MCF-7), as well as after pleural effusion initiation four years after mastectomy (MDA-MB231) [11,12]. Despite many advantages of conventional 2D in vitro cell cultures such as being well-known, cost-effective and easy to operate, they have significant limitations. First of all, they do not reflect the complexity of in vivo structures because they lack 3D architecture, cell-to-cell interactions, and communication between different tumor cell types [13]. In commercial immortalized cell lines, genetic aberrations accumulate with each passage, limiting their usefulness and affecting the final screening. Moreover, they do not allow for capturing differences in patients’ responses to the same drugs used to treat cancers with identical genetic mutations [14].

Currently, short-term cell culture of primary solid tumors is gaining more and more importance in personalized cancer therapy. Primary cultures maintain the stem-like phenotype of tumor cells, therefore ex vivo models allow for the accurate representation of tumors and are more suitable for clinical analysis, unlike the most commonly used immortalized cell lines, which may not fully predict tumor expression [14].

Thus, commercial cell lines do not retain the cellular diversity found in human breast tissues [15]. The results of Weigand et al. indicate that only assemblies of a few cell lines representing multiple cell types can be used to study the response to the proposed chemotherapy [16].

In our study, we confirmed the presence of cancer cells by assessing the expression of a group of CD44(+) and CD24(+) receptors. Although we are aware that in the collected material there were also cells that did not demonstrate this expression, as they belonged to other cell lines. We have proven that it is possible to maintain a cell culture obtained from a biopsy for three months. This supports the optimization of the conditions for culturing cells isolated from the tumor. We used an IMDM medium, which is a highly enriched medium well suited for rapidly proliferating, high-density cell cultures. It probably contributed to the breeding success. Cytometric analysis performed three months after collecting the material confirmed the presence of three cellular subpopulations (Appendix A). It cannot be ruled out that the presence of non-cancerous cells contributed to the increased survival of the cancer cells, which made it possible to maintain the culture for three months after isolation. In addition, daily observation of the cells under an inverted microscope showed that the death of the non-cancerous cells was rapidly followed by the death of the cancerous cells. We are aware that long-term cell culture leads to genetic and non-genetic evolution and is also exposed to mycoplasma infection [17].

A detailed analysis of patients 4 and 7 showed some similarity in the expression levels of GATA-3, cytokeratin, and Ki-67 (Figure 4A,C,E). However, there were statistically significant differences between these patients. Confirmation of the correctness of the cytometric analysis is the result of a histochemical examination performed within two weeks of the biopsy, in which the cytometric trend is maintained. Some differences between immunohistochemistry and cytometry prove that certain changes occur within three cell cultures within three months—the time of maintaining cell lines and performing cytometric analysis. The results of cytometric as well as immunohistochemical tests confirm that the cells isolated from the biopsy retain their properties for the test period—three months. Therefore, to reduce the time necessary to offer oncologists drugs to which the patient’s cells are sensitive, and thus with a high potential for effectiveness, only one of these methods could be used. However, we suggest that this test should be performed only in situations of uncertainty when the results of other tests (MTT test, cell morphology, LDH activity assessment) make it impossible to make these proper decisions. The evaluation of extracellular and intracellular protein expression was aimed only at confirming that cancer cells were present in the examined “cellular cocktail”. We are aware that sometimes it may happen that the drugs selected in vitro cannot be used due to the patient’s clinical condition, e.g., low cardiac ejection fraction, or kidney or liver failure. In such situations, these studies will be of much-limited value.

The available literature includes studies describing the isolation of cancer cells from the tumor [18], as well as assessing the effect of several anticancer compounds on commercial cancer lines [19]. The advantage of our research is the fact that the material for it was isolated from a specific patient and, as part of personalized medicine, the optimal therapy is selected for a specific patient. In our research, we used a set of anticancer drugs used in the chemotherapy ward. Therefore, the exposure of cancer cells to currently used anticancer drugs in our in vitro model realistically reflects the conditions occurring in the body of a breast cancer patient undergoing chemotherapy.

The proposed procedure creates a real chance for the effective elimination of cancer cells in every patient, demonstrating high effectiveness, which in turn translates into the possibility of implementing this method into routine therapy of breast cancer patients. Another very valuable element of the diagnostic and therapeutic procedure described above is the reduction of the side effects observed during chemotherapy, which does not always prove to be effective. Our procedure eliminates the possibility of using therapy to which the cancer cells are not sensitive, which gives a chance to use effective chemotherapy from the first days. The way of proceeding proposed by our team, due to its high applicability, is undoubtedly an attractive offer for therapeutic entities or pharmaceutical companies both in Poland and around the world. In the future, it may find application in oncology, in the treatment of not only breast cancer but also other types of solid tumors.

The limitations of this study are due to the fact that the proposed diagnostic and therapeutic pathway has so far been studied only in a small number of breast cancer patients. The study was conducted on a small group of only 8 patients above 60 years old. In the case of two patients (no. 4 and 7), detailed tests were performed, confirmed by an extensive research panel. However, in the case of the remaining six patients (no. 1, 2, 3, 5, 6 and 8), a shortened testing model was used. This research is of a pilot nature. The next stage will be research conducted on a larger number of patients, of different ages, with different histopathological types of breast cancer. The effectiveness of the present invention should be assessed on a much larger number of patients in the future. Some difficulties may be caused by slow-growing tumors, in the case of which it will be necessary to extend the diagnostic and therapeutic period, as well as situations in which tumor cells are not isolated from the biopsy. However, our experience shows that we can obtain cancer cells in over 90% of cases.

## 5. Conclusions

Based on the study, it can be concluded that to select effective drugs, it is sufficient to perform MTT assay, morphological evaluation of tumor cells, and LDH activity in culture medium after exposure of tumor cells to drugs. This approach will reduce the test time and cost. The proposed diagnostic and therapeutic scheme supports precision medicine, contributing to the development of a personalized treatment plan tailored to individual needs, which leads to improved survival rates.

## 6. Patents

The results were submitted in the patent application No. P.444462.

## Figures and Tables

**Figure 1 cancers-16-00120-f001:**
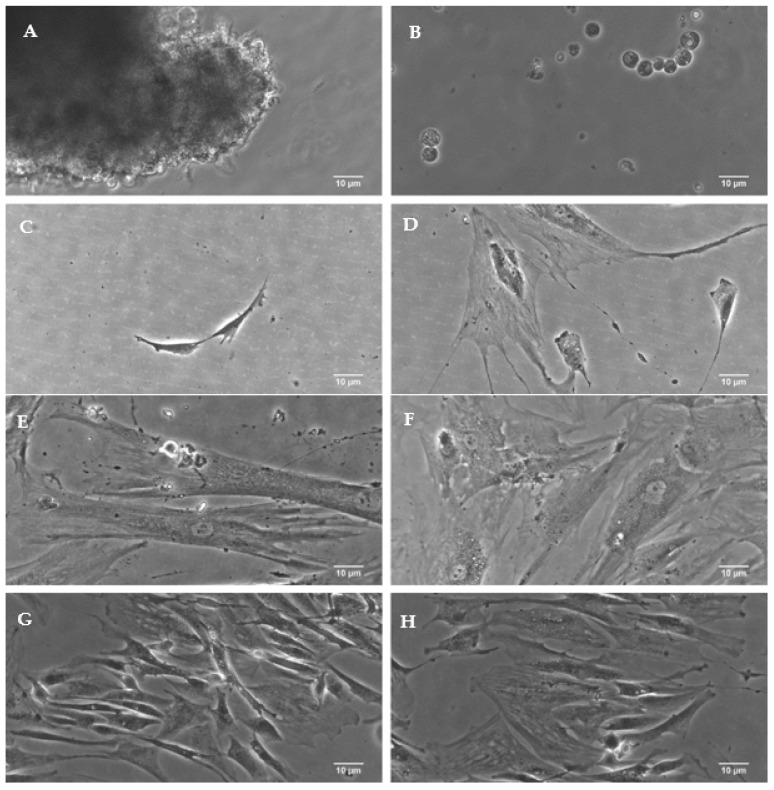
The stages of isolating cancer cells from a biopsy (**A**–**D**) and the successive stages of growth and proliferation of cells make up the adhering line (**E**–**H**).

**Figure 2 cancers-16-00120-f002:**
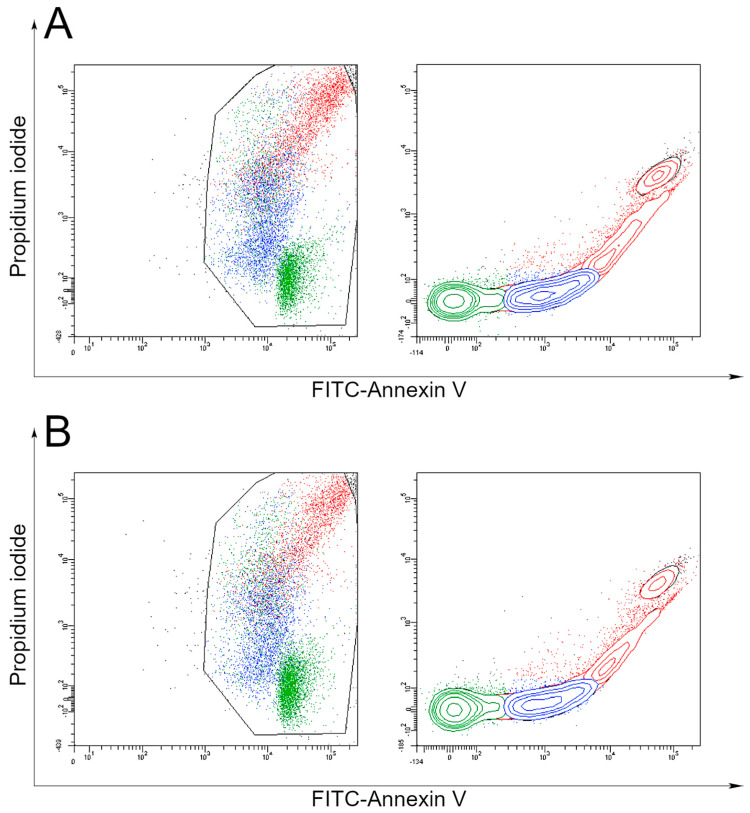
Confirmation of the preservation of cellular heterogeneity in the material obtained from the biopsy of patients no. 4 (**A**) and no. 7 (**B**). Three colors indicate the presence of three cell populations.

**Figure 3 cancers-16-00120-f003:**
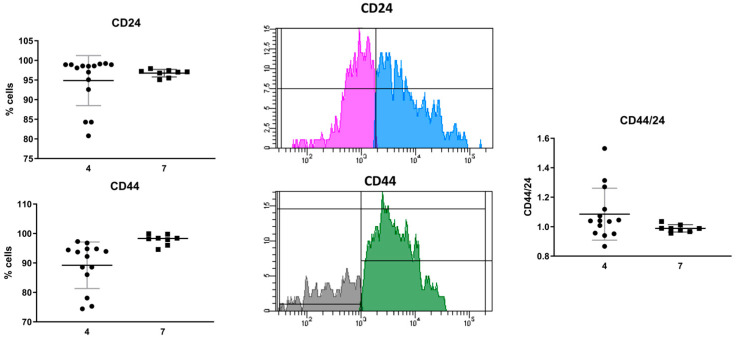
Evaluation of CD24 and CD44 receptor expression on cells isolated from the biopsy; sample cytogram, comparison of CD24 and CD44 receptor expression and CD44/24 expression ratio in patients no. 4 and no. 7.

**Figure 4 cancers-16-00120-f004:**
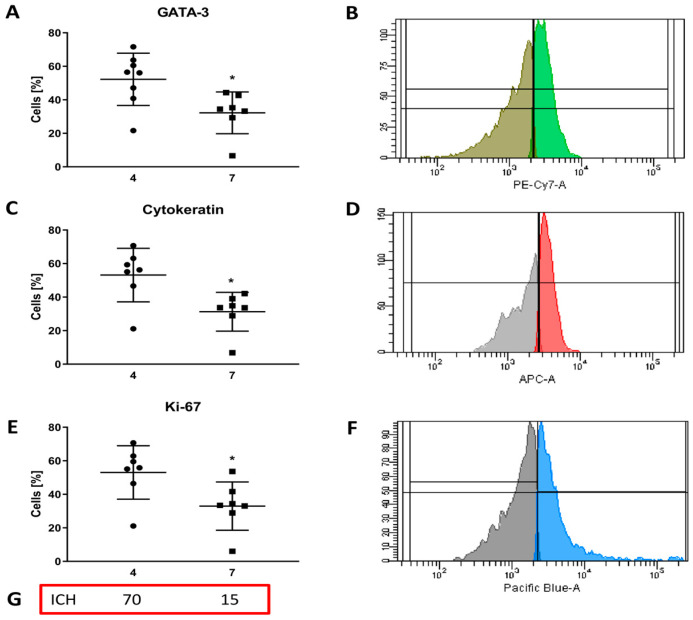
Expression of intracellular markers in cells obtained from the biopsy: GATA-3 (**A**) comparison of GATA-3 expression in patients no. 4 and no. 7; *p* = 0.018, (**B**) representative FACS plots of GATA-3; cytokeratin (**C**) comparison of cytokeratin expression in patients 4 and 7; *p* = 0.013, (**D**) representative FACS plots of cytokeratin; Ki-67 (**E**) comparison of Ki-67 expression in patients 4 and 7; *p* = 0.026, (**F**) representative FACS plots of Ki-67, (**G**) immunohistochemical results performed in November 2022, while cytometric analysis was performed in January 2023.

**Figure 5 cancers-16-00120-f005:**
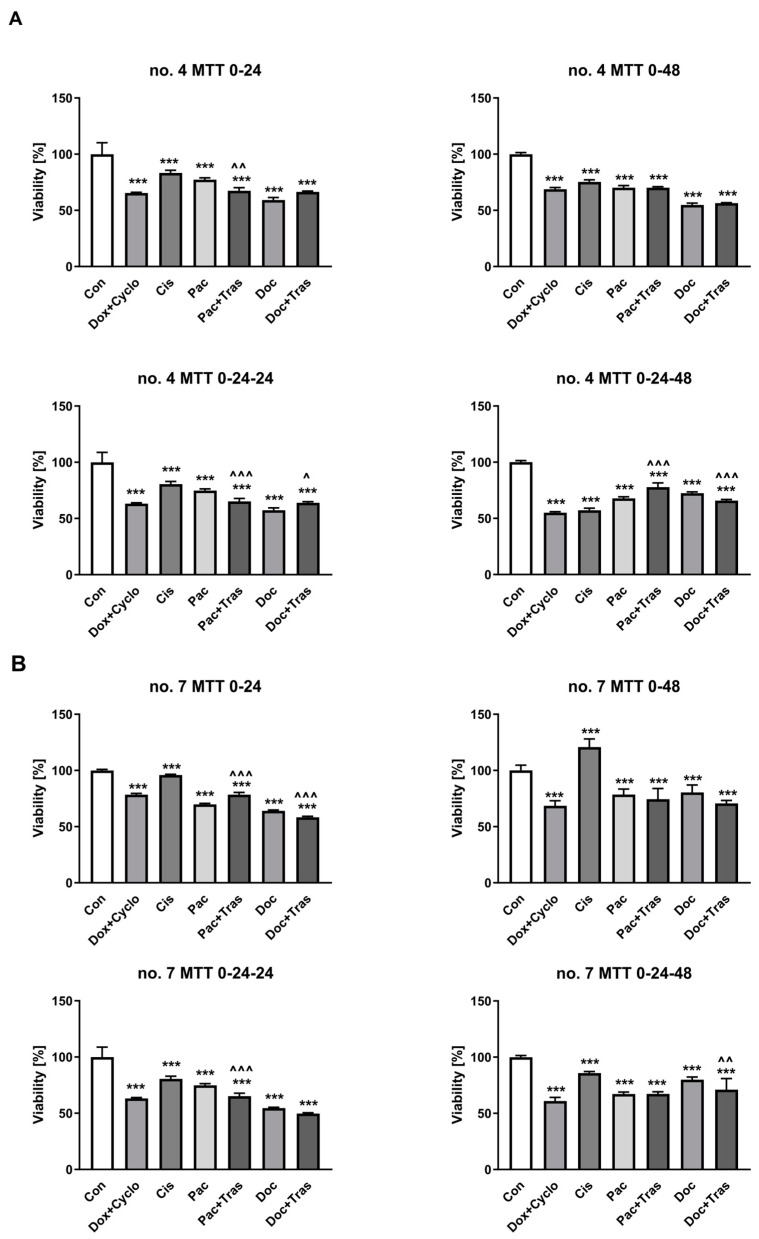
Evaluation of cell viability obtained from the biopsy after 24 and 48 h of incubation with drugs, patients no. 4 (**A**) and 7 (**B**). *** *p* < 0.0001 vs. Con, ^^ *p* < 0.001 vs. Doc, ^^^ *p* < 0.0001 vs. Pac; n = 6. 0–24 means that the cells immediately after sieving were treated with the compounds for 24 h; 0–24–24 means that the cells only 24 h after sieving were treated with the compounds for another 24 h; 0–48 means that the cells immediately after sieving were treated with the compounds and incubated with them for 48 h; 0–24–48 means that the cells only 24 h after sieving were treated with the compounds and incubated with them for 48 h; asterisks symbols mean comparison vs. Con, ^ means comparison vs. Pac or Doc.

**Figure 6 cancers-16-00120-f006:**
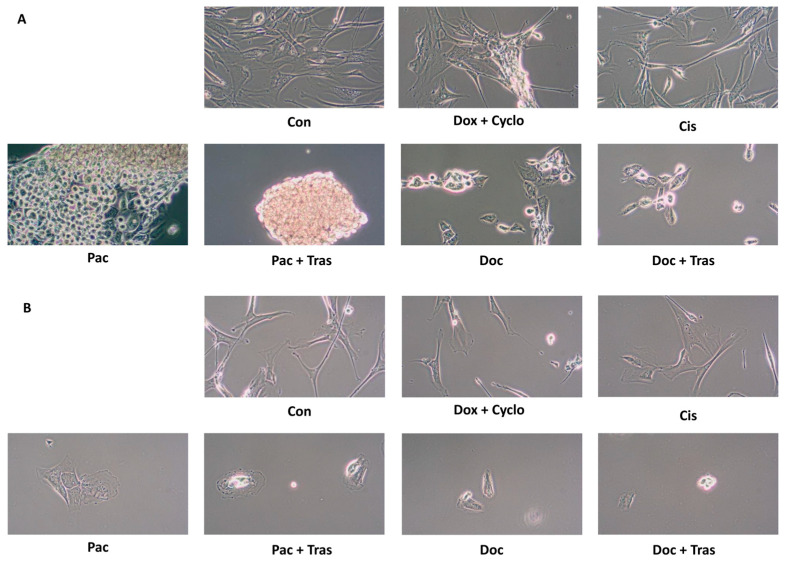
Morphometric evaluation of cells (**A**) patient no. 4, (**B**) patient no. 7 (×10) after 24 h incubation with drugs.

**Figure 7 cancers-16-00120-f007:**
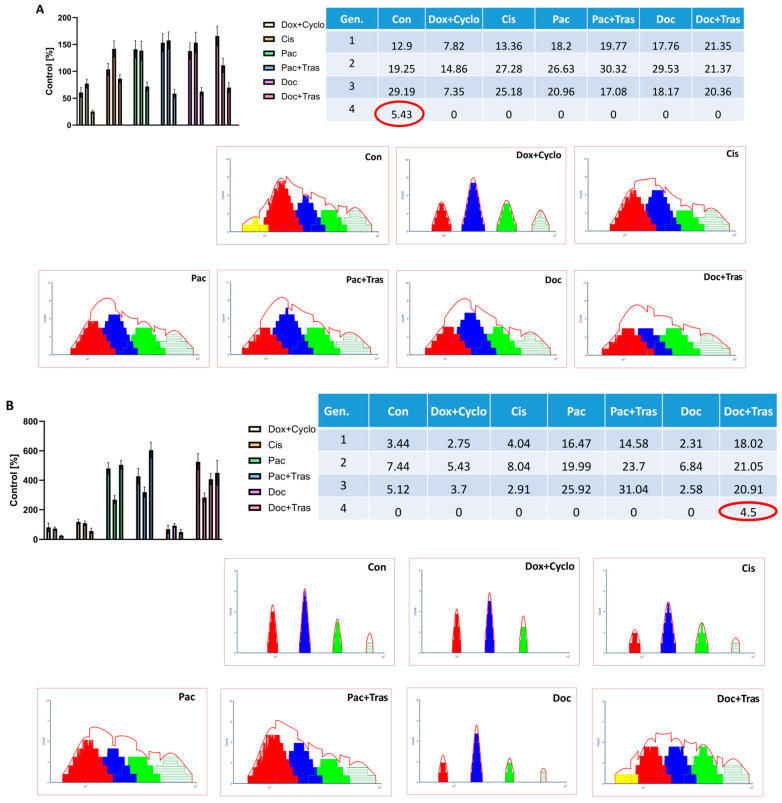
Evaluation of proliferation in patients no. 4 (**A**) and 7 (**B**). Red circles indicate cases in which the fourth generation of cells occurred, i.e., situations in which cells were still proliferating. In the case of patient 4, this situation occurred only in the control group not treated with drugs. However, in the case of patient 7, the use of Doc + Tras did not effectively inhibit the proliferation of cancer cells, as evidenced by the appearance of the 4th cell generation.

**Figure 8 cancers-16-00120-f008:**
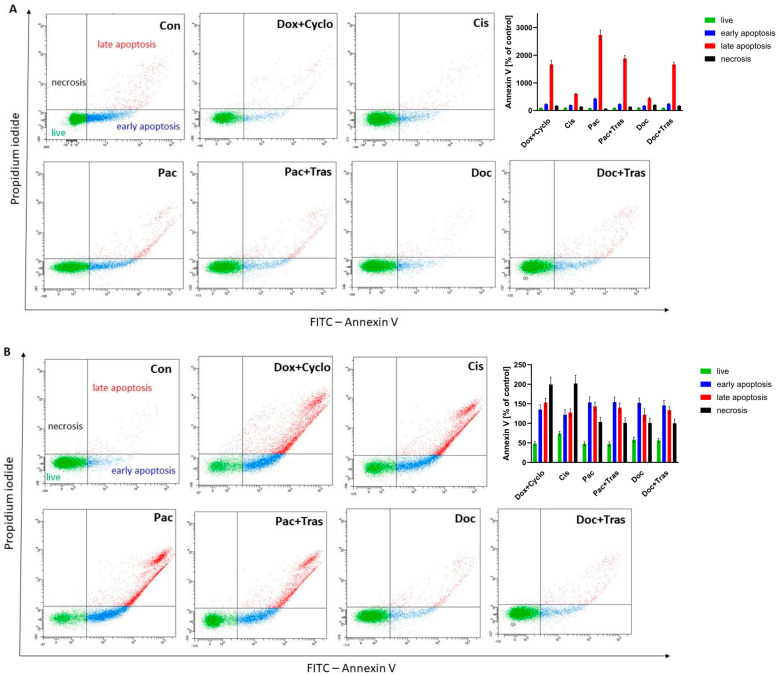
Identification of apoptosis by the FCM assay of annexin V/PI double staining. (**A**) The percentage of apoptotic cancer cells in patient no. 4 after 24 h incubation with drugs and representative flow cytometry dot plots for annexin V–FITC (fluorescein isothiocyanate) assay. (**B**) The percentage of apoptotic cancer cells of patient no. 7 after 24 h incubation with drugs and representative flow cytometry dot plots for annexin V–FITC (fluorescein isothiocyanate) assay.

**Figure 9 cancers-16-00120-f009:**
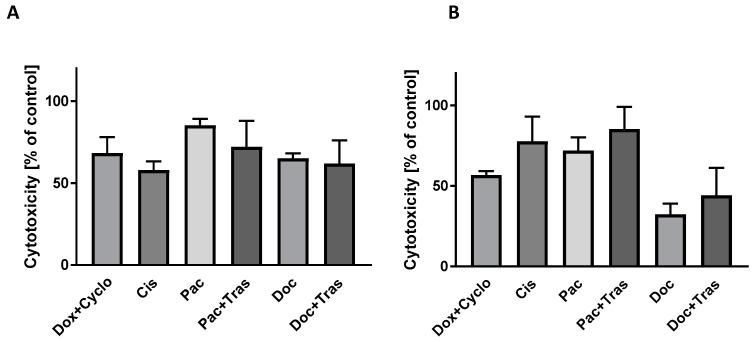
Evaluation of lactate dehydrogenase (LDH) activity in the culture medium after 24 h incubation of cancer cells from patients 4 (**A**) and 7 (**B**) with drugs.

**Table 1 cancers-16-00120-t001:** General characteristics of the patients. Patients subjected to detailed analysis are marked in red.

No.	Patient	Age	H-*p*	ER	PR	Ki-67	HER2
**1**	BM	54	NOS G3	0	5	60	0
**2**	IG	85	NOS G3	100	100	70	3+
**3**	AS	53	NOS G1	100	90	20	1+
**4**	IH	66	NOS G3	90	5	70	3+
**5**	DC	74	NOS G2	90	0	60	0
**6**	WZ	74	NOS G2	90	20	70	1+
**7**	CS	80	NOS G1	100	70	15	0
**8**	MM	49	NOS G1	80	100	10	2+

**Table 2 cancers-16-00120-t002:** Cell viability (MTT) assay after 24 h incubation with anticancer drugs.

Patient No.	Con	Dox + Cyclo	Cis	Pac	Pac + Tras	Doc	Doc + Tras
1	100 ± 2.34	89.00 ± 6.25 *p* = 0.0231	92.23 ± 4.60 *p* = 0.0288	82.50 ± 7.13 *p* = 0.0001	93.08 ± 2.51	86.13 ± 4.13 *p* = 0.0023	94.08 ± 3.48
2	100 ± 3.44	62.71 ± 5.24 *p* = 0.0001	28.00 ± 5.65 *p* = 0.0001	36.27 ± 9.79 *p* = 0.0001	21.06 ± 7.91 *p* = 0.0001	62.29 ± 10.64 *p* = 0.0001	64.58 ± 15.69 *p* = 0.0001
3	100 ± 8.43	73.88 ± 6.05 *p* < 0.0001	131.80 ± 12.14 *p* = 0.0259	113.15 ± 14.62	100.31 ± 35.36	105.81 ± 22.37	129.36 ± 13.47 *p* = 0.0447
5	100 ± 8.36	76.19 ± 1.45 *p* < 0.0001	102.18 ± 8.76	89.96 ± 6.71	91.09 ± 3.50	92.20 ± 3.38	80.06 ± 2.04 *p* < 0.0001 *p* = 0.0040 *
6	100 ± 9.22	65.86 ± 6.09 *p* = 0.0001	91.67 ± 8.44	79.03 ± 15.61 *p* = 0.0001	84.07 ± 5.40 *p* = 0.0006	66.67 ± 7.32 *p* = 0.0001	75.60 ± 1.92 *p* = 0.0001
8	100 ± 6.50	76.17 ± 7.05 *p* = 0.0001	93.06 ± 7.20	98.66 ± 5.65	89.80 ± 7.57 *p* = 0.231	76.96 ± 6.50 *p* = 0.0001	89.65 ± 7.01 *p* = 0.0268 *p* = 0.0280 *

Results are presented as mean ± SD, n = 3–6; * vs. Doc.

**Table 3 cancers-16-00120-t003:** Morphometric evaluation of cells after 24 h incubation with anticancer drugs.

Con	Dox + Cyclo	Cis	Pac	Pac + Trans	Doc	Doc + Trans
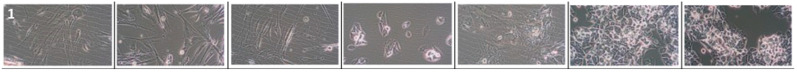
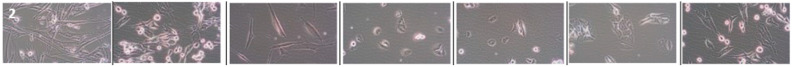
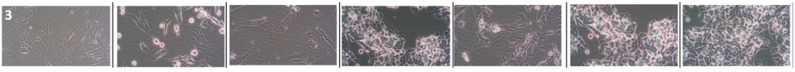
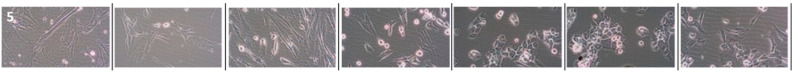
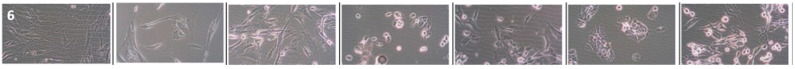
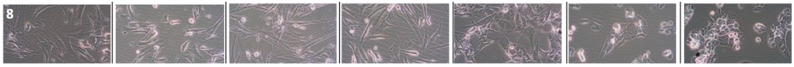

The numbers indicate the patient number.

**Table 4 cancers-16-00120-t004:** LDH activity in culture medium after 24 h incubation with anticancer drugs.

Patient	Con	Dox + Cyclo	Cis	Pac	Pac + Tras	Doc	Doc + Tras
1	3.53 ± 1.22	26.83 ± 5.04 *p* = 0.0410	10.13 ± 4.03	23.10 ± 5.40	20.93 ± 7.136	26.13 ± 8.07	22.93 ± 4.98
2	4.57 ± 0.65	41.8 ± 9.50 *p* = 0.0006	76.03 ± 6.89 *p* = 0.0001	52.27 ± 15.10 *p* = 0.0001	68.13 ± 6.82 *p* = 0.0001	56.73 ± 7.75 *p* = 0.0001	57.27 ± 6.87 *p* = 0.0001
3	3.60 ± 1.10	33.37 ± 12.22 *p* = 0.0001	6.90 ± 3.90	11.40 ± 2.86	11.13 ± 4.00	13.27 ± 4.99	6.80 ± 4.83
5	4.83 ± 0.91	34.10 ± 7.42 *p* = 0.0002	13.27 ± 7.20	17.17 ± 7.50	11.07 ± 5.48	14.80 ± 5.62	24.90 ± 5.83
6	3.46 ± 0.60	46.37 ± 6.79 *p* = 0.0001	14.67 ± 5.75	24.30 ± 6.85 *p* = 0.0033	26.63 ± 7.51 *p* = 0.0013	36.17 ± 6.25 *p* = 0.0001	19.63 ± 4.35 *p* = 0.0210
8	4.40 ± 0.60	39.13 ± 6.26 *p* = 0.0001	10.40 ± 1.91	11.87 ± 3.07	15.00 ± 5.85	28.80 ± 8.44 *p* = 0.0002	19.80 ± 4.37 *p* = 0.0103

Results are presented as mean ± SD, n = 3–6.

**Table 5 cancers-16-00120-t005:** Drug selection criteria based on chemosensitivity assessment.

No.	MTT	Morphology	LDH	Suggested Chemotherapy
1	Dox + Cyclo, Cis, Pac, Doc	Dox + Cyclo, Cis, Pac	Dox + Cyclo	Dox + Cyclo
2	Dox + Cyclo, Cis, Pac, Pac + Tras, Doc, Doc + Tras	Dox + Cyclo, Cis, Pac, Pac + Tras, Doc, Doc + Tras	Dox + Cyclo, Cis, Pac, Pac + Tras, Doc, Doc + Tras	Dox + Cyclo, Cis, Pac, Pac + Tras, Doc, Doc + Tras
3	Dox + Cyclo	Dox + Cyclo	Dox + Cyclo	Dox + Cyclo
4	Doc, Doc + Tras, Dox + Cyclo	Pac, Pac + Tras, Doc, Doc + Tras	Pac, Pac + Tras, Doc + Cyclo, Doc + Tras	Doc + Tras
5	Dox + Cyclo	Dox + Cyclo, Cis	Dox + Cyclo	Dox + Cyclo
6	Dox + Cyclo, Pac, Pac + Tras, Doc, Doc + Tras	Dox + Cyclo, Pac, Pac + Tras	Dox + Cyclo, Pac, Pac + Tras, Doc, Doc + Tras	Dox + Cyclo, Pac, Pac + Tras
7	Dox + Cyclo, Pac, Pac + Tras, Doc + Tras	Pac, Pac + Trass, Doc, Doc + Tras	Pac, Pac + Tras, Cis	Pac, Pac + Tras
8	Dox + Cyclo, Pac + Tras, Doc, Doc + Tras	Dox + Cyclo, Doc, Doc + Tras	Dox + Cyclo, Doc, Doc + Tras	Dox + Cyclo, Doc, Doc + Tras

## Data Availability

The data may be requested from the corresponding author. All requests will be checked according to privacy and possible ethical restrictions.

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
