# Peer review of "ONCOBREAST-TEST Is a Quick Diagnostic, Prognostic and Predictive Method of Response to Systemic Treatment"

_cancers, 2023, doi:10.3390/cancers16010120_

Round 1
Reviewer 1 Report
Comments and Suggestions for Authors
Dear author’s
I was pleased to review your article and i have the following comment’s:
The subject is interesting. Please specify in the abstract a short sentence about methods and results.
Please specify if there are studies with this hypothesis.
There are some differences betwen different type of breast cancer?
The study sample is too small and for this reason is very difficult to draw conclusions.
Please highlight the clinical relevance of this test.
Minor English edits.
Author Response
We would like to take this opportunity to deeply thank the Reviewer who identified the parts of our manuscript that required corrections or modifications. Please find the response to the Rewiever’s comments below.
Please specify in the abstract a short sentence about methods and results.
Changes in the manuscript in the Abstract section:
Page 1, lines 29-30
Chemosensitivity of cancer cells was assessed using the MTT test, morphological assessment of cells, LDH activity in the culture medium and flow cytometry technique (apoptosis, proliferation, CD24, CD44, GATA3, cytokeratin, Ki-67).
Page 1, lines 35-37
The proposed procedure allows for a simple (based on MTT results, cell morphology, LDH concentration), minimally invasive, quick and accurate assessment of the sensitivity of breast cancer cells to the drugs used and, to select the most effective treatment plan as part of personalized therapy. In a patient with NOS G3 the most promising therapy will be docetaxel with cyclophosphamide and in the case of a patient with NOS G1 paclitaxel alone and in combination with trastuzumab.
Please specify if there are studies with this hypothesis.
The available literature includes studies describing the isolation of cancer cells from the tumor (Pershina O, Ermakova N, Pakhomova A, Widera D, Pan E, Zhukova M, Slonimskaya E, Morozov SG, Kubatiev A, Dygai A, Skurikhin EG. Cancer Stem Cells and Somatic Stem Cells as Potential New Drug Targets, Prognosis Markers, and Therapy Efficacy Predictors in Breast Cancer Treatment. Biomedicines. 2021 Sep 14;9(9):1223), as well as assessing the effect of several anticancer compounds on commercial cancer lines (Zhao M, Lei C, Yang Y, Bu X, Ma H, Gong H, Liu J, Fang X, Hu Z, Fang Q. Abraxane, the Nanoparticle Formulation of Paclitaxel Can Induce Drug Resistance by Up-Regulation of P-gp. PLoS One. 2015 Jul 16;10(7):e0131429). The advantage of our research is the fact that the material for it was isolated from a specific patient and, as part of personalized medicine, the optimal therapy is selected for a specific patient. In our research, we used a set of anticancer drugs used in the chemotherapy ward. Therefore, the exposure of cancer cells to currently used anticancer drugs in our in vitro model realistically reflects the conditions occurring in the body of a breast cancer patient undergoing chemotherapy.
Changes in the manuscript in the Discussion section:
Page 20, lines 598-605
The available literature includes studies describing the isolation of cancer cells from the tumor [17], as well as assessing the effect of several anticancer compounds on commercial cancer lines [18]. The advantage of our research is the fact that the material for it was isolated from a specific patient and, as part of personalized medicine, the optimal therapy is selected for a specific patient. In our research, we used a set of anticancer drugs used in the chemotherapy ward. Therefore, the exposure of cancer cells to currently used anticancer drugs in our in vitro model realistically reflects the conditions occurring in the body of a breast cancer patient undergoing chemotherapy.
There are some differences betwen different type of breast cancer?
Changes in the manuscript in the Results section in 3.1. General Patient and Tumor Characteristics
Page 5, lines 231-243
The study included 8 patients with a diagnosis of invasive breast carcinoma (IBC), not otherwise specified (NOS) (Invasive ductal carcinoma) in different grades of histologic differentiation (G1-3). NOS G1 type occurred in 3 patients (37.5% of cases), NOS G2 type occurred in 2 patients (25% of cases) and NOS G3 type occurred in 3 patients (37.5% of cases). IBC NOS is the most common type of invasive breast carcinoma (75%-80%). All IBC NOS arise from epithelial progenitor cells at the terminal duct lobular unit in the mammary gland. Histologic grade (G) is independent and one of the most important prognostic factors in breast carcinoma. Pathologists use the Nottingham, modified Bloom & Richardson Score for diagnosis [7]. Histologic grading is based on microscopic analysis of tubule formation (1 - 3 points), nuclear pleomorphism (1 - 3 points) and mitotic count (1 - 3 points). Total score (after adding points for tubule formation, nuclear pleomorphism and mitotic count) divide all IBC NOS on: grade 1 (G1; 3 - 5 points), grade 2 (G2; 6 - 7 points) and grade 3 (G2; 8 - 9 points).
The study sample is too small and for this reason is very difficult to draw conclusions.
Indeed, these studies were conducted on a small group of only 8 patients. In the case of two patients (no. 4 and 7), detailed tests were performed, confirmed by an extensive research panel. However, in the case of the remaining six patients (no. 1, 2, 3, 5, 6 and 8), a shortened testing model was used. This research is of a pilot nature. The next stage will be research conducted on a larger number of patients, of different ages, with different histopathological types of breast cancer. We fully agree with the Reviewer's suggestion. However, we believe that preliminary conclusions can already be drawn at this stage of the study. And we hope that future research will provide confirming evidence.
Changes in the manuscript in the Discussion section:
Page 20, lines 619-624
The study was conducted on a small group of only 8 patients above 60 years old. In the case of two patients (no. 4 and 7), detailed tests were performed, confirmed by an extensive research panel. However, in the case of the remaining six patients (no. 1, 2, 3, 5, 6 and 8), a shortened testing model was used. This research is of a pilot nature. The next stage will be research conducted on a larger number of patients, of different ages, with different histopathological types of breast cancer.
Please highlight the clinical relevance of this test.
We have presented a diagnostic and therapeutic procedure enabling the selection of appropriate chemotherapy dedicated to a specific oncological patient. Its undoubted advantage is its simplicity, sensitivity and cheapness.
The proposed procedure guarantees the effective elimination of cancer cells in each patient, demonstrating high effectiveness, which in turn translates into the possibility of implementing this method into the routine therapy of patients with breast cancer.
Another very valuable element of the diagnostic and therapeutic procedure described above is the reduction of side effects observed during chemotherapy, which does not always turn out to be effective. Our approach eliminates the possibility of using therapy to which cancer cells are not sensitive, which gives us a chance to use effective chemotherapy from the first days.
ONCOBREAST-TEST, due to its high applicability, is undoubtedly an attractive offer for medical entities and pharmaceutical concerns both in the country and around the world. In the future, it may find application in oncology, in the treatment of not only breast cancer but also other types of solid cancers.
The clinical significance of our research is presented in the Discussion section
Page 20, lines 606-619
The proposed procedure creates a real chance for the effective elimination of cancer cells in every patient, demonstrating high effectiveness, which in turn translates into the possibility of implementing this method into routine therapy of breast cancer patients. Another very valuable element of the diagnostic and therapeutic procedure described above is the reduction of the side effects observed during chemotherapy, which does not always prove to be effective. Our procedure eliminates the possibility of using therapy to which the cancer cells are not sensitive, which gives a chance to use effective chemotherapy from the first days. The way of proceeding proposed by our team, due to its high applicability, is undoubtedly an attractive offer for therapeutic entities or pharmaceutical companies both in Poland and around the world. In the future, it may find application in oncology, in the treatment of not only breast cancer but also other types of solid tumors.
Minor English edits.
The test was corrected by a native speaker.

Reviewer 2 Report
Comments and Suggestions for Authors
The ONCOBREAST-TEST proposes a personalized approach for selecting anti-cancer drugs in breast cancer. Utilizing simple and cost-effective tests like cell viability assessment, morphology, and lactate dehydrogenase activity, the procedure aims to identify drugs with optimal efficacy against a patient's cancer cells obtained from a biopsy. In vitro studies demonstrate a high likelihood of accurate drug selection using these methods. The work is very interesting and could have great potential Some points are listed below for the author's consideration:
- Statistical Analysis:
- While the results mention statistically significant differences, the absence of specific statistical values (e.g., p-values, confidence intervals) diminishes the robustness of the findings. Incorporating these details would provide a clearer understanding of the significance of the observed differences. Include specific statistical values in the results section. This could involve integrating p-values or confidence intervals to quantify the significance of observed differences.
- Confounding Variables:
- The section lacks discussion on potential confounding variables that might influence the results. Factors such as patient age, tumor stage, or treatment history could impact the outcomes. Addressing these variables explicitly will enhance the validity of the study.
- Cell Line Diversity:
- The discussion on the limitations of commercial cell lines is valid, but there's a need to incorporate recent studies that delve deeper into the diversity of cancer cell lines. Highlighting the limitations of monoculture in comparison to more diverse representations would strengthen the argument. Strengthen the discussion on cell line diversity by referencing recent studies that provide comprehensive insights into the limitations of monoculture. Elaborate on the significance of maintaining cell cultures for three months. Reference relevant literature to support the stability and reliability of long-term cultures
- Long-Term Viability:
- While the study mentions maintaining the cell culture for three months, there's a lack of discussion on the implications of this duration. Address the potential changes in cell behavior over time and reference studies that explore the stability of cancer cell cultures for extended periods.
- Comparison with Existing Methods:
- The results could be enriched by comparing the proposed diagnostic scheme with existing methods in more detail. Provide insights into how the proposed approach outperforms or complements current diagnostic strategies, adding depth to the discussion.
Author Response
We would like to take this opportunity to deeply thank the Reviewer who identified the parts of our manuscript that required corrections or modifications. Please find the response to the Rewiever’s comments below.
- Statistical Analysis:
- While the results mention statistically significant differences, the absence of specific statistical values (e.g., p-values, confidence intervals) diminishes the robustness of the findings. Incorporating these details would provide a clearer understanding of the significance of the observed differences. Include specific statistical values in the results section. This could involve integrating p-values or confidence intervals to quantify the significance of observed differences.
Thank you for this suggestion. The confidence intervals were added to 3.3.2. Evaluation of Extracellular Marker Expression, 3.3.3. Evaluation of Intracellular Marker Expression, 3.4.1. Cell Viability Testing, 3.4.3. Assessment of Cell Proliferation, 3.4.4. Evaluation of Apoptosis and 3.4.5. Evaluation of LDH Activity in Culture Medium sections.
Detailed data are provided in Tables S1-S6 in the Supplementary.
- Confounding Variables:
- The section lacks discussion on potential confounding variables that might influence the results. Factors such as patient age, tumor stage, or treatment history could impact the outcomes. Addressing these variables explicitly will enhance the validity of the study.
The study included women over 60 years of age who were diagnosed with breast cancer based on the H-P test from biopsy material. These patients did not receive anticancer treatment, but due to age and concomitant diseases, such as hypertension, they received convertase inhibitors and β-blockers. However, the therapy used so far did not affect the chemosensitivity of cancer cells.
Changes in the manuscript in the Materials and Methods section, subsection 2.1. Study Population:
Page 2, lines 76-78
These patients did not receive anticancer treatment, but due to age and concomitant diseases, such as hypertension, they received convertase inhibitors and β-blockers.
- Cell Line Diversity:
- The discussion on the limitations of commercial cell lines is valid, but there's a need to incorporate recent studies that delve deeper into the diversity of cancer cell lines. Highlighting the limitations of monoculture in comparison to more diverse representations would strengthen the argument. Strengthen the discussion on cell line diversity by referencing recent studies that provide comprehensive insights into the limitations of monoculture. Elaborate on the significance of maintaining cell cultures for three months. Reference relevant literature to support the stability and reliability of long-term cultures
Following the Reviewer's suggestion, we have expanded the Discussion section.
Changes in the manuscript in the Discussion section:
Page 19, lines 546-559
Despite many advantages of conventional 2D in vitro cell cultures such as being well-known, cost-effective and easy to operate, they have significant limitations. First of all, they do not reflect the complexity of in vivo structures because they lack 3D architecture, cell-to-cell interactions, and communication between different tumor cell types [12]. In commercial immortalized cell lines, genetic aberrations accumulate with each passage, limiting their usefulness and affecting the final screening. Moreover, they do not allow for capturing differences in patients' responses to the same drugs used to treat cancers with identical genetic mutations [13].
Currently, short-term cell culture of primary solid tumors is gaining more and more importance in personalized cancer therapy. Primary cultures maintain the stem-like phenotype of tumor cells, therefore ex vivo models allow accurate representation of tumors and are more suitable for clinical analysis, unlike the most commonly used immortalized cell lines, which may not fully predict tumor expression [13].
- Long-Term Viability:
- While the study mentions maintaining the cell culture for three months, there's a lack of discussion on the implications of this duration. Address the potential changes in cell behavior over time and reference studies that explore the stability of cancer cell cultures for extended periods.
Changes in the manuscript in the Discussion section:
Page 19, lines 567-572
We have proven that it is possible to maintain a cell culture obtained from a biopsy for three months. This supports the optimization of the conditions for culturing cells isolated from the tumor. We used IMDM medium, which is a highly enriched medium well suited for rapidly proliferating, high-density cell cultures. It probably contributed to the breeding success. Cytometric analysis performed three months after collecting the material confirmed the presence of three cellular subpopulations (Figure S1).
Page 19, lines 577-578
We are aware that long-term cell culture leads to genetic and non-genetic evolution [6] and is also exposed to mycoplasma infection.
- Comparison with Existing Methods:
- The results could be enriched by comparing the proposed diagnostic scheme with existing methods in more detail. Provide insights into how the proposed approach outperforms or complements current diagnostic strategies, adding depth to the discussion.
Thank you for this suggestion. Following the Reviewer's suggestion, we have expanded the Discussion section.
Changes in the manuscript in the Discussion section:
Page 20, lines 598-605
The available literature includes studies describing the isolation of cancer cells from the tumor [17], as well as assessing the effect of several anticancer compounds on commercial cancer lines [18]. The advantage of our research is the fact that the material for it was isolated from a specific patient and, as part of personalized medicine, the optimal therapy is selected for a specific patient. In our research, we used a set of anti-cancer drugs used in the chemotherapy ward. Therefore, the exposure of cancer cells to currently used anticancer drugs in our in vitro model realistically reflects the conditions occurring in the body of a breast cancer patient undergoing chemotherapy.
However, our goal is not to expand the research panel to include molecular tests, expression of selected genes, or omics studies, but to select, in the shortest possible time, the drug(s) to which the cancer cells are most sensitive.

Reviewer 3 Report
Comments and Suggestions for Authors
Authors have aimed at addressing the necessity for personalized medicine by proposing a simple diagnostic and therapeutic assay, ONCOBREAST-TEST, which aims to select drugs against which a patient's cancer cells taken during a biopsy show the greatest sensitivity. The research has attempted to show that simple, inexpensive and quick tests such as the assessment of cell viability, morphology, and lactate dehydrogenase activity are sufficient tools for selecting drugs with the highest anticancer efficacy.
1. The patient group is very small, and limited to women above 60 years old. This should be more emphasized as the substantial limitations of the study.
2. Fig. 3 - how can be differences in the measurements of CD24- and CD44-positive cells for patient 4 explained? Similarly, why so huge dispersion was observed between the results for a single patient's sample? This suggests technical problems with staining.
3. Legend for Fig. 7 is insufficient. Why were certain values red circled?
4. Fig. 8 - the graphs lack S.D. Why were the results shown as % of control (i.e. as relative values)?
5. The results shown in Fig. 9 show that LDH level is reduced upon treatment, while data presented in Table 4 show the opposite effect.
6. The majority of figures is hardly legible and look like they were cut and pasted from other software.
7. Overall, the quality of the data and their interpretation poorly support the conclusion that "The proposed procedure creates a real chance for the effective elimination of cancer cells in every patient, demonstrating high effectiveness, which in turn translates into the possibility of implementing this method into routine therapy of breast cancer patients." The study is too preliminary, and validated using only two random samples.
Comments on the Quality of English Languageminor revision required
Author Response
We would like to take this opportunity to deeply thank the Reviewer who identified the parts of our manuscript that required corrections or modifications. Please find the response to the Rewiever’s comments below.
- The patient group is very small, and limited to women above 60 years old. This should be more emphasized as the substantial limitations of the study.
Thank you for this suggestion. These are pilot studies, but the results obtained raise great hopes in our opinion.
Changes in the manuscript in the Discussion section:
Page 20, lines 619-624
The study was conducted on a small group of only 8 patients above 60 years old. In the case of two patients (no. 4 and 7), detailed tests were performed, confirmed by an extensive research panel. However, in the case of the remaining six patients (no. 1, 2, 3, 5, 6 and 8), a shortened testing model was used. This research is of a pilot nature. The next stage will be research conducted on a larger number of patients, of different ages, with different histopathological types of breast cancer.
- Fig. 3 - how can be differences in the measurements of CD24- and CD44-positive cells for patient 4 explained? Similarly, why so huge dispersion was observed between the results for a single patient's sample? This suggests technical problems with staining.
Page 9, lines 309-312
It is difficult to explain why such large changes were observed in the patient. Perhaps they were dictated by the heterogeneity of cell populations. CD24 and CD44 expression results are presented in scatter dot plots as means and SD. Originally they were presented in the form of box and whiskers (min to max).
- Legend for Fig. 7 is insufficient. Why were certain values red circled?
Indeed, the caption of Figure 7 requires refinement. Thank you for this pertinent comment. Red indicates cases in which the 4th generation of cells occurred, i.e. situations in which cells were still proliferating. In the case of patient no. 4, this situation occurred only in the control group not treated with drugs. However, in the case of patient no. 7, the use of Doc+Tras did not effectively inhibit the proliferation of cancer cells, as evidenced by the appearance of the 4th cell generation.
The caption of Figure 7 has been enriched with the following sentences:
Page 13, lines 410-414
Red indicates cases in which the 4th generation of cells occurred, i.e. situations in which cells were still proliferating. In the case of patient no. 4, this situation occurred only in the control group not treated with drugs. However, in the case of patient no. 7, the use of Doc+Tras did not effectively inhibit the proliferation of cancer cells, as evidenced by the appearance of the 4th cell generation.
- Fig. 8 - the graphs lack S.D. Why were the results shown as % of control (i.e. as relative values)?
Following the Reviewer's suggestion, S.D. was added to Figure 8 (and also to Figure 7). The purpose of presenting the results as % of control was to minimize the number of columns and thus achieve better readability. Also, the results showing LDH concentration were expressed as % changes compared to the control group. In the case of LDH concentration determination, a positive control in which all cells are lysed (LDH concentration is the highest) had to be used, as well as a negative control with the lowest LDH concentration. The concentration in the control group was calculated using the formula.
Cytotoxicity (%) = exp. value - low control/ high control - low control x 100
Also in this case, we wanted to display the results in the study groups while ensuring maximum readability of the figures. To maintain the same trend in the presentation of results, proliferation and apoptosis were also presented as % changes compared to the control.
- The results shown in Fig. 9 show that LDH level is reduced upon treatment, while data presented in Table 4 show the opposite effect.
Figure 9 shows the LDH concentration in the culture medium in patients no. 4 and 7. Table 4 shows the results of tests on patients no. 1, 2, 3, 5, 6 and 8. Because these were different patients with different types of cancer we observed different responses to treatment, including different LDH concentrations.
- The majority of figures is hardly legible and look like they were cut and pasted from other software.
Following the Reviewer's suggestion, the figures have been redacted.
- Overall, the quality of the data and their interpretation poorly support the conclusion that "The proposed procedure creates a real chance for the effective elimination of cancer cells in every patient, demonstrating high effectiveness, which in turn translates into the possibility of implementing this method into routine therapy of breast cancer patients." The study is too preliminary, and validated using only two random samples.
Indeed, these studies were conducted on a small group of only 8 patients. In the case of two patients (no. 4 and 7), detailed tests were performed, confirmed by an extensive research panel. However, in the case of the remaining six patients (no. 1, 2, 3, 5, 6 and 8), a shortened testing model was used. This research is of a pilot nature. The next stage will be research conducted on a larger number of patients, of different ages, with different histopathological types of breast cancer. We fully agree with the Reviewer's suggestion. However, we believe that preliminary conclusions can already be drawn at this stage of the study. And we hope that future research will provide confirming evidence.
Changes in the manuscript in the Discussion section:
Page 20, lines 619-624
The study was conducted on a small group of only 8 patients above 60 years old. In the case of two patients (no. 4 and 7), detailed tests were performed, confirmed by an extensive research panel. However, in the case of the remaining six patients (no. 1, 2, 3, 5, 6 and 8), a shortened testing model was used. This research is of a pilot nature. The next stage will be research conducted on a larger number of patients, of different ages, with different histopathological types of breast cancer.

Round 2
Reviewer 1 Report
Comments and Suggestions for Authors
Thank you for your revised manuscript.
Author Response
We would like to take this opportunity to deeply thank the Reviewer who identified the parts of our manuscript that required corrections or modifications.
Reviewer 3 Report
Comments and Suggestions for Authors
The manuscript has been substantially improved.
Author Response

(The authors gave the same response as above.)
